# Estimating CO_2_ Emissions from IoT Traffic Flow Sensors and Reconstruction

**DOI:** 10.3390/s22093382

**Published:** 2022-04-28

**Authors:** Stefano Bilotta, Paolo Nesi

**Affiliations:** DISIT Lab, Department of Information Engineering, University of Florence, 50139 Firenze, Italy; stefano.bilotta@unifi.it

**Keywords:** smart city, vehicle CO_2_ emissions factor, traffic flow, reconstruction algorithm, traffic congestion, regression CO_2_ model, seasonal changing

## Abstract

CO_2_ emissions from burning fossil fuels make a relevant contribution to atmospheric changes and climate disruptions. In cities, the contribution by traffic of CO_2_ is very relevant, and the general CO_2_ estimation can be computed (i) on the basis of the fuel transformation in energy using several factors and efficiency aspects of engines and (ii) by taking into account the weight moved, distance, time, and emissions factor of each specific vehicle. Those approaches are unsuitable for understanding the impact of vehicles on CO_2_ in cities since vehicles produce CO_2_ depending on their specific efficiency, producer, fuel, weight, driver style, road conditions, seasons, etc. Thanks to today’s technologies, it is possible to collect real-time traffic data to obtain useful information that can be used to monitor changes in carbon emissions. The research presented in this paper studied the cause of CO_2_ emissions in the air with respect to different traffic conditions. In particular, we propose a model and approach to assess CO_2_ emissions on the basis of traffic flow data taking into account uncongested and congested conditions. These traffic situations contribute differently to the amount of CO_2_ in the atmosphere, providing a different emissions factor. The solution was validated in urban conditions of Florence city, where the amount of CO_2_ is measured by sensors at a few points where more than 100 traffic flow sensors are present (data accessible on the Snap4City platform). The solution allowed for the estimation of CO_2_ from traffic flow, estimating the changes in the emissions factor on the basis of the seasons and in terms of precision. The identified model and solution allowed the city’s distribution of CO_2_ to be computed.

## 1. Introduction

Traffic assessment and control is a central topic for intelligent transportation systems (ITS). Thanks to today’s technologies, real-time data can be collected and used to monitor and control vehicular traffic. The knowledge of real-time traffic information enables the development of a relevant number of services and improvements in many areas: congestion detection and reduction, dynamic network traffic control, improved information services (e.g., traffic information, dynamic route guidance, road digital signage), planning for future investments on mobility solutions, and reducing fuel consumption and pollution emissions. Recently, regarding the latter, there has been a deeper understanding of the environmental parameters (for example, PM10 (particulate matter), PM2.5, CO (carbon monoxide), CO_2_ (carbon dioxide), SO_2_ (sulphur dioxide), O_3_ (ozone), H_2_S (hydrogen sulphide), NO (nitric oxide), NO_2_ (nitric dioxide), and NOx (nitric monoxide and dioxide)) and how much they are influenced by a city’s structures, what the causes of high values of pollutants are, the reasons for the registered high values from the IOT Data Network, and the dynamic of their diffusion and propagation [1,2]. For more on this, see the normative of the European Commission regarding the conformant of the environmental value with respect to the reference values (2008/50/EC Directive on Ambient Air Quality and Cleaner Air for Europe and 2004/107/EC Directive on Heavy Metals and Polycyclic Aromatic Hydrocarbons in Ambient Air). Greenhouse gas (GHG) emissions make a relevant contribution to atmospheric changes and climate disruptions. Many gases contribute to global warming, and the CO_2_ emissions from burning fossil fuel represents the primary contribution in the transportation field. Traffic emissions from fuel combusted in vehicles are typically estimated by multiplying activity data by emission factors (EFs) [3]. For example, the distance travelled by a vehicle has a large influence on emissions, since in general more activity leads to greater emissions. Vehicle speed is another important influence on emissions, because road traffic EFs are strongly dependent on speed [4,5]. Emissions also depend on vehicle category. Different vehicle categories have different EFs due to factors such as vehicle mass, fuel specification, engine size, aerodynamics, and emissions control technology. One of the most commonly used methodologies for fleet-wide emissions assessment is the emissions inventorying methodology included in the EMEP/EEA guidebook [6]. A specific report on car emissions of CO_2_ can be found in [7], where traffic emissions of CO_2_ is analysed, identifying the typical emissions factors for different kinds of vehicles and their trends over time. In that context, different emissions factors have been identified according to fuel type, efficiency, producer, power, weight, etc. Therefore, almost all CO_2_ emissions in vehicular traffic depend on a variety of vehicle- and traffic-related parameters, such as vehicle characteristics and motorisation, driving behaviour, and traffic conditions. Thus, most of the approaches for measuring CO_2_ are based on (i) energy models that take into account the conversion of total consumed energy by vehicles, taking into account efficiency, carbon content per unit of heat, carbon oxidation factor, and rate [8], or on (ii) transport models that take into account the average distance, the emissions factor of the vehicles, and the weight moved [7]. 

### 1.1. Literature Review

The present research studies the cause of CO_2_ emissions in the air, which are primarily related to the impact of vehicular traffic emissions in uncongested and congested situations, assuming that the population of the vehicle type depends on the city area but is stationary over time. Roughly speaking, traffic congestion is the deterioration of smooth free-flowing traffic conditions due to increased travel demand and/or reduced traffic movement capacity [9]. It is commonly accepted that under the so-called “stop-and-go” traffic conditions associated with congestion there is an increase in the number of accelerations and decelerations performed by vehicles, which result in increased emissions [10,11]. The study in [12] measured instantaneous speed, acceleration, travel time, fuel consumption, and exhaust emissions in a field test to find that fuel consumption depends largely on traffic flow patterns and traffic conditions. Fuel consumption under congested flow conditions is higher than under free-flow conditions due to the frequency of operations such as acceleration, deceleration, and “stop-and-go” driving. Various tools exist for calculating CO_2_ emissions and simulating vehicle operations on a vehicle-specific level (micro-level). For example, the European Union recently introduced a vehicle simulation model, CO_2_MPAS, for CO_2_ certification purposes [13]. Other works combined VISSIM traffic simulation software with different fuel consumption and emissions models to study vehicle fuel consumption and emissions at a given simulated signalized intersection [14,15]. In [16], a model for estimating the amount CO_2_ in grams (briefly ${\mathrm{gCO}}_{2}$ from traffic has been proposed. The model assessed a mean emissions factor of 203 ${\mathrm{gCO}}_{2}/\mathrm{km}$ per vehicle for an “urban slow” pattern, for “*stop-and-go A*” pattern a mean emissions factor of 460 ${\mathrm{gCO}}_{2}/\mathrm{km}$ per vehicle, and for “*stop-and-go B*” an emissions factor of 738 ${\mathrm{gCO}}_{2}/\mathrm{km}$ per vehicle. The driving modalities assumed that when the number of cars is at a minimum there is no traffic congestion and the urban slow pattern can be considered. When the traffic corresponds to the mean observed value, the “*stop-and-go A*” model is applied, and when the traffic reaches its maximum highest emissions factor, the “*stop-and-go B*” pattern can be used. In [17], an approach to annual road traffic CO_2_ emissions estimation for an urban area with street-level resolution was conducted. In the mentioned study, the CO_2_ emissions factors were computed by applying a TRL (Transport Research Lab) dataset [18] to the fleet vehicle composition in an urban area, and the related modelling uncertainties, running from 6.5% to 12%, focused on peak traffic periods, when significant emissions of CO_2_ occur.

In order to control and reduce emissions, national and international organisations have defined guidelines and targeted limits to be respected currently, and to be progressively reduced over the years/months. In this regard, the European Union has set limits for the yearly mean value of the concentration of pollution. In [19], a model and tool to compute long-term predictions was proposed, up to 180 days in advance, for the progressive mean value of NO_2_ with a precision needed to enable decision-makers to perform corrections. In this context, the TRAFAIR project (Understanding Traffic Flows to Improve Air Quality) has been focused on the short- and mid-term prediction of NOx on the basis of traffic flow emissions [20]. 

### 1.2. Paper Aim and Structure

In this paper, we present a *model to estimate the CO_2_ emissions from traffic flow* data, characterising the specific city traffic flow in terms of emissions factor, which is regarded as the amount of CO_2_ produced per vehicle for unit distance (gCO_2_/km per car). As reported in the above analysis, the literature on the state of the art estimates the CO_2_ in (i) controlled experiments for each single vehicle type, as in [12,13,14,15], and (ii) a general approach on the basis of the energy consumed (fuel) by taking into account of vehicle distribution, as in [16,17]. Please note that a global measurement of CO_2_ could be performed by satellites, but those measurements are affected by multiple problems, since they (1) cannot be precisely attributed to a specific city area since they perform the measurement on the basis of the column of air from the ground to satellite quote, and (2) are imprecise in the presence of clouds, wind, air humidity, etc. The proposed method, unlike the approaches presented in [12,13,14,15,16,17], allows the CO_2_ emissions to be estimated on city roads. In particular, the proposed method: Allows the CO_2_ produced by vehicles in a city to be estimated, which is a relevant contribution of CO_2_ produced in cities, directly measuring the impact of vehicle population on the production of CO_2_, thus increasing the precision in measuring CO_2_. The advantage of identifying the function from traffic flow to CO_2_ is that it can be used to estimate the CO_2_ in city areas based on knowing the traffic flow with a certain precision, and thus the total CO_2_ production of the city, which currently can only be coarsely guessed by using a very limited number of CO_2_ sensors, whereas most cities have hundreds of traffic flow sensors.Is based on (i) the identification of the relationships from the measured traffic flow and determining the emissions factors, taking into account different traffic behaviours, from fluid traffic to “stop-and-go” conditions (congested and uncongested traffic situations); (ii) the changes in the emissions factor at different periods of the year; (iii) an approach to statistical validation by means of the CO_2_ measurements taken from specific sensors and traffic flow data, allowing for an assessment of the precision of the indirect estimation of CO_2_ on the basis of traffic flow.Provides a solution for computing CO_2_ emissions locally in the city and not only for specific vehicles or globally at city level, as mentioned above in the literature review.

Moreover, in the validation approach, a number of problems have been solved, such as (a) the positions of traffic flow sensors typically not being aligned with those of air quality, (b) the measurements of CO_2_ not being temporally aligned with those of traffic flow sensors, and (c) different areas possibly impacting CO_2_ in a different manner depending on the city. The model presented in this paper and the related results were produced and validated by exploiting the Snap4City framework for smart cities, mobility and transport, and data analytics, also using Km4City/Sii-Mobility models and tools. The research was funded by the National Ministry of Research [21]. The algorithms were executed by exploiting the Snap4City data aggregator and its semantic model [22,23] and the Snap4city platform [24,25]. 

The paper is structured as follows and is depicted in Figure 1, which describes the data flows and algorithms put in place and the corresponding sections of the paper. In Section 2, we present the data description and the related problems. In Section 3, the analysis of traffic flow data is reported to pose the basis to identify the traffic flow modalities and needed normalisations. In Section 4, the model for the identification of congested and uncongested traffic flows is presented. In Section 5, the problems related to the different sensor locations for measuring CO_2_ and traffic flow data area addressed, presenting the traffic flow reconstruction algorithm to estimate traffic flow far from the sensor locations. Section 6 presents the model approach to compute the amount of CO_2_ in the air on the basis of the measured traffic flow in uncongested and congested situations in terms of emission factors. The experimental results (see Section 6.4) demonstrate (i) that the emission factors change in different periods of the year, and (ii) the precision of the indirect estimation of CO_2_ on the basis of traffic flow. Conclusions are drawn in Section 7.

## 2. Data Description and Related Problems

As mentioned in the introduction, the main goal of the present work is to find a relationship between vehicular traffic flow sensors and CO_2_, which can also be measured. Thus, the validation of the model could be viable in specific conditions, or a more general model has to be defined. Both air quality sensors and vehicular traffic sensors are taken into account for the present dissertation, and a description of the measurements that are typically performed is needed before discussing the model. Typically, **CO_2_ measurements** are performed either by the count of the particles in the air (commonly, *part per million, ppm*) or by means of the CO_2_ weight in a given air volume ($\mathrm{mg}/m^{3}$ or $g/m^{3}$) for certain temporal windows in a given location. In any city, the number of CO_2_ sensors is limited. For example, in Florence, which is a metro city area of 1.5 million inhabitants, about 10 sensors are present. However, in the downtown area, only four are present, denoted by their IDs: SMART09, SMART27, SMART28, and SMART29 (Figure 2a for the respective locations in the municipality of Florence, from [25]). Those sensors are not all in critical locations for traffic or for pollutants. SMART28 and SMART29 are in dense traffic roads, whereas SMAR27 and SMART09 are in mid-range traffic areas. For each air-quality sensor, the data are registered every 2 min. In most European cities the number of CO_2_ sensors is not much higher. Those sensors were chosen since they are in critical points of the city and they are located close to major roads. Therefore, it is reasonable to assume that the CO_2_ measured by a sensor represents the measurement taken in a given area around the sensor itself. They are calibrated in this sense. The presence of almost collocated CO_2_ and of traffic flow sensors allowed us to compute the emission factors as described in the following section. On the other hand, the absence or significant reduction of them could be overcome with the usage of rented CO_2_ sensors or by using the typical values of the emission factors in the area, diminishing the precision of the measurement.

Typically, traffic sensor data are simultaneously registered every 10 min, and their number is much higher than those of CO_2_ and in general for air quality assessments. The data exploited refer to (about) 100 traffic flow devices located in the municipality of Florence, as depicted on the right side of Figure 2b. They are spire and virtual spire sensors that produce values of counting, traffic density, and thus, local traffic flows. All of them are well calibrated and produce coherent results.

Analysing their geo distribution (see Figure 2a,b), it can be observed that the CO_2_ sensor locations have one or more traffic sensors in their proximity, but they are not precisely co-located. Thus, the actual traffic in the CO_2_ sensor position has to be estimated. To this end, the so-called technique for traffic flow reconstruction [26,27] may help in this sense, as described in the following section. The traffic flow measurements are strongly dependent on a number of road features: road relevance (primary, secondary, etc.), number of lanes, speed limits, presence of speed meters, distance from road crossings, etc. Moreover, a certain class of roads (e.g., the so-called primary/main roads) may provide higher capability with respect to local and single-lane cases. Traffic flow sensors provide at each time slot different measurements regarding vehicular traffic flow, such as:Vehicular traffic flow: number of vehicles crossing the supervised location during a given period of time (which is usually referred to in terms of hours, that is, #cars/h);Vehicular average speed: average speed of the vehicles crossing the supervised location (measured in km/h);Vehicular density: number of vehicles in terms of road occupancy (measured in #cars/km).Travel time: average time that vehicles take to transit the supervised area (reported in s).

Each traffic sensor datum may present a specific behaviour depending on the context. At each traffic sensor location, the measured vehicular traffic flow (density) depends on the number of lanes of the road where the traffic sensor is placed. In general, the measured vehicular traffic flow (density) coming from a traffic sensor located in a multiple-lane road is greater with respect to the traffic flow (density) value coming from a single-lane road. Moreover, the traffic sensors often consist of optical cameras, and the related measured data depend on specific parameters that are set up on the traffic sensor itself, that is, the selected area to be monitored and the length of the observed road segment. For example, when a traffic sensor is located on a long, straight road, it is set up in order to monitor a long road segment. However, when the traffic sensor is placed in the proximity of a curve in the road or a road junction, the supervised area is smaller. 

Each traffic sensor monitors a fixed supervised area that is constituted by means of a given road segment length. Then, each traffic sensor admits specific travel time measurements. In the absence of traffic congestion at the traffic sensor locations, the measured travel time (to cross the segment) is higher when the related monitored area is greater/longer, since the vehicles need more time to travel. More precisely, in the absence of traffic congestion, each traffic sensor admits a minimum (monitored) travel time, which can be defined as the vehicular time needed to across the supervised area within (at most) the speed limit occurring at the related traffic sensor location. Thus, different minimum cross/travel time can be registered for different traffic sensors. In order to compare the amount of traffic data coming from the whole urban network, different traffic behaviours were considered. For example, cumulative data were collected at several timesteps and the dataset was reordered according to increasing travel time. Figure 3 shows the monitored data trend of about 50 traffic sensors during seven days of observation, from Monday to Sunday (24 h for day), taking into account the relationship between the measured vehicular traffic flow with respect to the measured travel time.

When a large number of traffic sensors are taken into account in an urban road network, the complexity of the problem increases, and a uniform approach is needed in order to properly handle and compare the amount of traffic data at different traffic locations. On this basis, it can be supposed that the increment in travel time at a given point may be due to the inception of crowding conditions, and that not all sensor locations may be prone to crowding conditions. In the next section we introduce a uniform approach to formally understand when uncongested and congested situations occur in the road network by analysing a large amount of traffic data.

## 3. Traffic Flow Data Analysis

As reported in the introduction, the focus is to propose a general approach to determine a model to pass from traffic flow data to CO_2_ emissions, taking into account (i) a uniform approach that allows the measurements to be compared according to the physical context of sensor placement, (ii) the different traffic conditions (congested vs. uncongested), and (iii) seasonal changes. To achieve the proposed scope, a traffic data alignment is needed in order to handle the mentioned measurements by means of appropriate data normalisation. Since all the measured traffic data are affected by the physical context of the sensor placements, the traffic data must be compared in a specific way. In order to do that, the following normalisation approach was conducted for each *i*-th traffic sensor:Measured vehicular traffic flow, denoted by F(t) at a given timestamp t, is normalized with respect to the number of lanes, denoted by C, of the road of the location. Thus, the normalized vehicular traffic flow at a given timestamp t, denoted by Fn(t), is given by Fn(t)=F(t)/C;Measured travel time, denoted by T(t) at a given timestamp t, is normalized with respect to the *minimum travel time*, denoted by Tm, occurring in the absence of congestion in the sensor location. Tm can be defined as the travel time needed to cross the area at the speed limit of the observed segment. Thus, the normalized travel time at a given timestamp t, denoted by m(t), is given by m(t)=T(t)/Tm.

According to the above normalisation, the traffic flow data in Figure 3 can be compared as depicted in Figure 4. In this way, all the curves in Figure 4 started from the origin and the data were aligned. For example, consider two distinct road segments monitored by means of a given traffic sensor. Suppose that the first monitored road segment is a two-lane street 0.35 km long with a speed limit of 50 km/h and a minimum travel time of 25.2 s; the second monitored road segment is a single-lane street 0.25 km long with a speed limit of 30 km/h and a minimum travel time of 30 s. At a given time, the sensor located in the first monitored road segment registers 800 vehicles/h and the related travel time is 37.8 s. The sensor in the second monitored road segment registers 400 vehicles/h and the related travel time is 45 s. After the above normalisation, both the traffic situations in the example admit the same behaviour: 400 vehicles/h cross the single-lane street at 1.5 times their minimum travel times. The measurements are now aligned and comparable. The normalised measurements are represented by means of the same point/contribution in the traffic data alignment shown in Figure 4.

The **normalized travel time** is a dimensionless value denoted by *m*, and it can be considered a *multiplier factor* of a given minimum travel time. So, we set *m =* 1 in the absence of traffic congestion for each traffic sensor location and the related traffic data were comparable. 

In order to describe the traffic data in the whole network, the average traffic flow value was estimated with respect to the corresponding average travel time in the network. Since each travel time at a given location can be expressed in terms of multiplier factor of its minimum travel time, then an average value can be estimated by considering all the traffic sensor locations in the network. Figure 5 shows such average behaviour by considering the data in Figure 4. This approach allows the traffic flow to be characterized in the whole city in a given period of time (or time slot) in which the traffic flow data are collected.

Let us start to study the traffic conditions by considering the described mean traffic behaviour in order to observe uncongested and congested traffic situations. The vehicular traffic flow and travel time data are largely influenced by congested traffic conditions. More precisely, congested traffic situations have a higher vehicular traffic flow in the monitored road section. Moreover, the travel time depends on the vehicular average speed, which comes close to 0 when traffic congestion occurs. Higher travel time reduces the vehicular flow, and it is inversely proportional to the number of vehicles passing. The uncongested and congested traffic situations are implicitly determined by means of the volume of vehicular flow that passes the supervised area in the unit of time. To consider mean/typical behaviour of the whole network for a such value, we can take into account the average values of the normalised vehicular flow, denoted by Fn, and the normalised travel time m. The *vehicular flow rate* at time *t* can be defined as follows: $$FR\left(t\right)=\frac{Fn\left(t\right)}{m\left(t\right)}$$
which is, in terms of fluid dynamics, the volumetric flow rate [28] (also known as volume flow rate, rate of fluid flow), that is, the volume of a fluid passing in the time unit. Coming from this general definition, we introduced a similar concept in the traffic flow theory to estimate and observe the variation of traffic flow in terms of vehicular flow rate. For instance, the traffic situations described in the mentioned example after the normalisation approach admit the same contribution in terms of vehicular flow rate, which is equal to 266.6 vehicles/h. Such a normalisation is necessary to compare the flow rate in each data observation and observe the related variation. 

In Figure 6, the general behaviour of the *mean flow rate* (MFR) is presented by taking into account the data in Figure 5. It considers the (average) values of the normalised traffic flow and the corresponding normalised travel time in a given period of time, where the observations are sorted according to increasing travel time measurements in the whole network. The flow rate can also be computed through the data coming from a single traffic sensor in order to understand the variation in traffic modality at a specific location of the road network. Nevertheless, the described normalisation approach has been always conducted to compare different local traffic variations.

MFRs can immediately provide evidence of the whole network changing, and they allow different traffic behaviours to be determined according to the seasonal changes. In this work, the data taken into account were those from March, May, July, and October 2021 in order to analyse different situations in different seasons. More precisely, each season observation is represented by means of a period of 7 days, from Monday to Sunday, 24 h a day. Figure 7 shows the seasonal changes according to the *mean flow rate* by considering all of the traffic sensors in the network of Florence, where the related behaviours in March, May, July, and October are described by means of the curves coloured blue, green, yellow, and orange, respectively.

As expected, the general behaviour of the MFR in March is similar to the one in October, and the MFR in July is similar to the one in May. Of course, the meteorological conditions influence vehicle usage in terms of driving behaviour and volume. During the cold seasons the variation in traffic modality seems to be more emphasised with respect to the warm seasons, during which a relevant number of city users get around with motorbikes and bikes, which influence less or none the flow counting. Different seasonal behaviours can also be observed numerically by means of the standard deviation of each mentioned MFR, computed by $\sqrt{\frac{1}{N}\overset{N}{\underset{t=1}{\sum }}{\left(MFR\left(t\right)-\bar{MFR}\right)}^{2}}$, where *N* is the number of observations and $\bar{MFR}$ is the mean, which is equal to 134.52, 122.85, 114.55, and 135.92 for March, May, July, and October, respectively, according to the values in Figure 7. The greatest difference can be observed between the curves representing the behaviour in October and July, in orange and yellow, respectively, in Figure 7. Table 1 shows such a deviation in terms of absolute differences at some corresponding points. 

## 4. Traffic Flow Modalities: Congested and Uncongested

The present section is devoted to formally identifying traffic flow modalities: uncongested and congested traffic situations that occur, and when we can detect being in one of the two cases. The traffic flow data analysis reported in the previous section may allow us to understand when a congested situation occurs at a given traffic sensor location. In particular, by considering the MFR behaviour, two distinct situations in the diagrams were identified. From the behaviour of FR (which is absolutely similar to the trend of MFR), a change in concavity in the related diagram was identified. In particular, for each traffic sensor a specific FR trend was present, and there was a point in which the FR curve changed concavity. That point is the point in which the travel time started to decrease and the traffic started to congest. Such a concavity changing in each diagram can be determined by means of the unique inflection point, and the corresponding value in the flow rate can be identified when a reduction in the vehicles passage is taking place. More precisely, when the concavity of the flow rate is upwards, the vehicles passage assumes a quick increase and the traffic flow proceeds unimpeded. Then, the situation of uncongested traffic is assumed. Otherwise, when the concavity of the flow rate is downwards, the vehicle passage reduces to a quasi-constant condition. Such a traffic modality occurs when the flow is slowed down, and stop-and-go situations arise since the road capability is limited. Therefore, a congested traffic situation can be assumed for such behaviour. Such a concavity change may be determined by identifying the inflection point in the FR trend, as detailed in the following section. In order to formally estimate the inflection point, we analyse the function f, which is defined by means of a given FR behaviour. More precisely, each monitored traffic location provides an FR trend, which can be compared with other trends by performing the traffic data alignment described in the previous sections. Therefore, it is possible to model the FR trend with a function f—for example, for a time period of one week (7 days, from Monday to Sunday, 24 h a day, with traffic measurements every 10 min by the traffic sensors). In order to identify function f, we proceed to perform a polynomial approximation to minimise the worst-case error. Thus, the polynomial approximation P of function f minimises |P(x)−f(x)|, where x varies over the chosen interval. Therefore, the approximation P of function f is obtained by using a third-order polynomial form: $P\left(x\right)=ax^{3}+bx^{2}+cx+d$. Different values of the coefficients (a,b,c,d) of P determine a different traffic behaviour in terms of FR. For each *FR*(*T*,*i*) observed in a given period T at the *i*-th traffic location, a unique polynomial approximation is represented by $\left(a_{i}\left(T\right),b_{i}\left(T\right),c_{i}\left(T\right),d_{i}\left(T\right)\right)$. For example, by assuming the period T, running from 2021-05-03T00:00:00 to 2021-05-09T23:59:59, we obtained the characterisation of the polynomial approximation, as seen in Table 2.

As is well known, since the polynomial approximation P is differentiability class $C^{2}$ for each set of finite coefficients that differs from zero (P, its first derivative $P^{\prime }$, and its second derivative $P^{\prime \prime }$, exist and are continuous), the condition $P^{\prime \prime }=0$ can be used to find the desired inflection point in the considered interval of vehicular traffic behaviour. Thus, in a fixed period of observation, both at a given traffic sensor location and in the whole network (by assuming average traffic behaviour), it is possible to compute the inflection point that determines a change in the traffic modality from congested to uncongested. This allows the different contributions of traffic flow to be taken into account in terms of CO_2_. Then, it is possible to consider uncongested and congested situations, both at a given traffic location and in the whole network (see Table 3 for the identified inflection point according to the traffic sensors with the polynomial approximations in terms of flow rate listed in Table 2).

## 5. Traffic Flow Reconstruction into CO_2_ Sensor Locations 

In order to identify a model that can allow us to compute CO_2_ values on the basis of traffic flow data, the most effective approach could be to validate the model in specific points for long time windows. In [16], a general model for the whole city, estimating the amount CO_2_ from traffic in grams (briefly $gCO_{2}$), was proposed. The model assessed a mean emissions factor of 203 $gCO_{2}/km$ per vehicle for the “urban slow” pattern, for the “*stop-and-go A*” pattern a mean emission factor of 460 $gCO_{2}/km$ per vehicle, and for the “*stop-and-go B*” an emission factor of 738 $gCO_{2}/km$ per vehicle. The amount of CO_2_ measurements by an air quality sensor is related to the space/volume surrounding the location of the sensor itself. Therefore, the measured CO_2_ would depend on the amount of traffic flow passing in the road segment volume, which is of pertinence to the air-quality sensor. In the previous section, we demonstrated that the FR function presents two main modalities, also considering the MFR for the whole city. As highlighted in Section 2, locations in which CO_2_ sensors are located typically do not present corresponding traffic sensors in their precise proximity; they may be hundreds of meters away. Moreover, CO_2_ sensor SMART09 also does not present traffic sensors in the area. On the other hand, the traffic of the road segments closest to the CO_2_ sensor impacts the measured CO_2_ directly. A traffic reconstruction technique [26,27] can be used to estimate the traffic flow density in the road segments closer to the CO_2_ sensor over time. To this end, traffic flow reconstruction can be used. The exploitation of traffic flow data for CO_2_ estimation is addressed in Section 6.

Traffic flow reconstruction [26,27] is the process of producing a value of traffic density (flow)—e.g., vehicle per meter (vehicles per minute)—for each road (or road segment, or a large number of road segments) by starting from a limited number of traffic sensors measuring traffic density (flow) on the road. The measures of traffic density are typically obtained by stationary sensors in strategic positions. The problem of traffic flow reconstruction is regarded as the solution of the LWR (Lighthill–Whitham–Richards) model [29,30], which models the traffic density in terms of the partial differential equation (PDE). The solution of the LWR model is not a trivial matter for large networks due to its computational complexity and constraints [31,32,33]. The estimation of the traffic distribution on junctions plays a crucial role in the effectiveness of the LWR model application in a real context in order to have a complete description of the road segments composing the urban network in terms of traffic density. Traffic flow reconstruction is performed by solving a nonlinear model based on the conservation of vehicles, described by the following scalar hyperbolic conservation law. On a single road, we have:$$\frac{\partial \rho \left(t,x\right)}{\partial t}+\frac{\partial f\left(\rho \left(t,x\right)\right)}{\partial x}=0,$$
where ρ(t,x) is the traffic density of vehicles, which admits values from 0 to $\rho_{\max}$, where $\rho_{\max}$ > 0 is the maximum traffic density; the f(ρ(t,x)) function is the vehicular flow, which is defined by means of the product ρ(t,x) v(t,x), where v(t,x) is the vehicle speed, and the boundary conditions are $\rho \left(t,h\right)=\rho_{h}\left(t\right)$, $\rho \left(t,k\right)=\rho_{k}\left(t\right)$ and the initial values are $\rho \left(0,x\right)=\rho_{0}\left(x\right)$, where x ∈(h,k). In the case of first-order approximation, we assume that v(t,x) is a decreasing function, depending on the density, and then the corresponding flux is a concave function. Thus, we consider the local speed of the vehicles to be $v\left(\rho \right)=v_{\max}\left(1-\frac{\rho }{\rho_{\max}}\right)$ and then $f\left(\rho \right)=v_{\max}\left(1-\frac{\rho }{\rho_{\max}}\right)\rho$, where $v_{\max}$ is the speed limit on a given road segment (these assumptions are referred to in the literature as Greenshield’s Model). The solution is obtained by an iterative process at finite differences on the basis of the traffic flow data in the sensor points. For each timestep, the traffic flow reconstruction is performed by producing a value of traffic density in each city road segment of the graph, which are typically 20 mt. 

The accuracy of the described solution primarily depends on the computation of the so-called traffic distribution matrices (TDM), that is, the traffic flow distribution at junctions. In order to model the traffic distribution at junctions, a distribution matrix can be used to describe the percentage of vehicles leaving each outcoming road with respect to those entering each incoming road. Thus, the traffic distribution matrix is defined as $\mathrm{TDM}={\left\{w_{ji}\right\}}_{j=n+1,\ldots ,n+m,i=1,\ldots ,n}$, so $0<w_{ji}<1$ and $\overset{n+m}{\underset{j=n+1}{\sum }}w_{ji}=1$, for i=1,…,n and j=n+1,…,n+m, where $w_{ji}$ is the percentage of vehicles arriving from the i-th incoming road and taking the j-th outcoming road (assuming that, at each junction, the incoming flux coincides with the outcoming flux). The real values of $w_{ji}$ may depend on the time of the day, the road size, the crossing light settings, etc., and thus, are unknown a priori. The values of $w_{ji}$ are estimated by giving the lower mean error by means of the stochastic relaxation technique as described in [26]. The traffic flow reconstruction algorithm has to be computed progressively and in a parallel architecture, since the estimation of traffic flow density for the city (in Florence there are about 30,000 segments) at time instant t would depend on traffic flow at time *t* − 1 and on the new measurements coming from the sensors. For each traffic sensor update, we have a complete description of the road segments composing the urban network in terms of traffic density.

## 6. Computing CO_2_ Emissions Factors from Traffic Flow Data and Modalities

In this section, we are going to identify how uncongested and congested traffic situations each contribute to the amount of CO_2_ measured. The amount of CO_2_ measured by a sensor is related to the area surrounding the sensor. The idea is to consider the amount of traffic contained in the road segments in which the CO_2_ sensors are located. The traffic flow reconstruction algorithm allows the amount of traffic to be computed in terms of vehicular density in each road segment with a length of 20 m every 10 min. By choosing, from the total number of reconstructed road segments, the closest one to each air-quality sensor, we can analyse the impact of traffic on the CO_2_ measurements in different zones of the city. In substance, we identified a model (see Section 6, Equation (1)) with which the computation of CO_2_ from traffic would be possible. The computation of CO_2_ depends on the traffic flow and of the emissions factors, which are different for congested and uncongested traffic flow cases. Therefore, we found a way to detect the conditions to compute the number of vehicles passing close to the CO_2_ control point in the different conditions (Section 4). This approach allows us to demonstrate (validate) the model by computing the emissions factors.

The measurements of CO_2_ and traffic flow refer to different systems of measurement, different time intervals, and different acquisition methods.

### 6.1. Time Alignment of Traffic Flow and Measured CO_2_ Data

Traffic flow data are available every 10 min, and CO_2_ data every 2 min. In order to align and compare the data coming from the air-quality sensors with the traffic data sensors, the CO_2_ measurements were carried over to the timesteps of the traffic sensors. Thus, the average of the CO_2_ data over the interval considered by the traffic sensors was taken into account (see the left side of Figure 8). Moreover, the standard traffic flow measurements are expressed in terms of #cars/h. Thus, we need to divide by 6 to get #cars/10 min; see an example on the right side of Figure 8. 

### 6.2. Pollutant Data Type

CO_2_ measurements are typically estimated either by the percentage of its particles in the total amount of the gasses in the air (commonly, part per million) or by means of the CO_2_ weight in a given volume of air (commonly, $mg/m^{3}$ or $g/m^{3}$). These standard units of measurement admit a direct conversion of one into the other and vice-versa, taking into account certain parameters such as the mole of the particulate matter under consideration. 

### 6.3. From Traffic Flow Data to CO_2_

In order to related the amount of CO_2_ in a given road segment with the vehicle density in its proximity, it is necessary to associate the CO_2_ measurement with a volumetric section of road segment. The measured emissions of CO_2_ in the volume should be equivalent to the CO_2_ produced by vehicles according to the traffic behaviour, which can be identified according to what was presented in Section 4. Therefore, we can assume to have two different contributions of emissions according to a classification of traffic behaviours. Thus, the following equation holds:(1)$$S\left(z\right)G\left(t,z\right)=K_{1}\left(z\right)F_{1}\left(t,z\right)L\left(z\right)+K_{2}\left(z\right)\;F_{2}\left(t,z\right)L\left(z\right)\;$$
where t is the time interval and z is the CO_2_ sensor ID.

S(z)G(t,z) is the amount of ${\mathrm{gCO}}_{2}$ in a volumetric section of the road segment at a given time *t*, where:
○G(t,z) is the measurement of CO_2_ from the sensor in ${\mathrm{gCO}}_{2}/m^{3}$ in the time interval (these values are measured by the CO_2_ sensors);○S(z) is the area in which the sensor collects the values in $m^{3}$ and it is estimated on the road segments close to the CO_2_ sensor location. More precisely, we have:(2)S(z)=L(z) C(z) W(z) H(z)
where C(z) is the number of lanes, W(z) is the width of the road lane, and H(z) is the height of the volume, which depends on the position of the CO_2_ sensor (typically at 3 m). ○L(z) is the road length corresponding to the amount of m or Km performed by the vehicles in that specific area of the CO_2_ sensor, supposing that the vehicles change neither road nor behaviour in the segment. Contribution coming from vehicles/cars moving in uncongested conditions:
○$F_{1}\left(t,z\right)$ is the traffic count in uncongested conditions in terms of #cars in the time interval. This can be measured on the basis of traffic sensors and/or estimated as solutions of the above-presented LWR PDE via the traffic reconstruction model (Section 5);○$K_{1}\left(z\right)$ is an emissions factor to be determined, which is the amount of gCO_2_/km per car in uncongested conditions. Contribution coming from vehicles/cars moving in congested conditions:
○$F_{2}\left(t,z\right)$ is the traffic count in congested conditions in terms of #cars in the time interval. This can be measured on the basis of traffic sensors and/or estimated as solutions of the above-presented LWR PDE via the traffic reconstruction model);○$K_{2}\left(z\right)$ is an emissions factor to be determined, which is the amount of gCO_2_/km per car in congested conditions. 

In Equation (1), for a certain time interval and sensors (CO_2_ and traffic), $K_{1}\left(z\right)$ and $K_{2}\left(z\right)$ are unknowns. They can be estimated by means of a multilinear regression providing a large number of measurements for both CO_2_ and traffic in the same contextual conditions: location, traffic, season, etc. Once $K_{1}\left(z\right)$ and $K_{2}\left(z\right)$ are determined, it is possible to apply Equation (1) to estimate CO_2_ emissions (i.e., G(t,z)) in other parts of the city provided that the other factors are known. Among them are the typical traffic behaviour, the distribution of vehicle types, etc., which can be estimated to be very similar in the same city and in closed areas/roads. 

Moreover, in Equation (1), $F_{1}\left(t,z\right)$ and $F_{2}\left(t,z\right)$ are obtained by classifying the vehicles passing in the road segment closest to the *z*-th CO_2_ sensor location according to the general condition of congested or uncongested. Thus, they are computed as the product between the traffic density (#cars/km) and the vehicular speed (km per time interval):(3a)$$F_{1}\left(t,z\right)=D_{1}\left(t,z\right)\;V_{1}\left(z\right)$$
(3b)$$F_{2}\left(t,z\right)=D_{2}\left(t,z\right)\;V_{2}\left(z\right)$$According to the following cases of congested and uncongested:If $\frac{\rho \left(t,z\right)}{C\left(z\right)}\leq q\left(z\right)$, then $D_{1}\left(t,z\right)=\rho \left(t,z\right)$ and $D_{2}\left(t,z\right)=0$;If $\frac{\rho \left(t,z\right)}{C\left(z\right)}>q\left(z\right)$, then $D_{1}\left(t,z\right)=0$ and $D_{2}\left(t,z\right)=\rho \left(t,z\right)$.where: ρ(t,z) is the traffic density (#cars/km) as observed on the traffic sensors and/or estimated as solutions of the above-presented LWR PDE via traffic flow reconstruction;q(z) is the value on the inflection point of traffic density in the proximity of the *z*-th air-quality sensor location to detect congested and uncongested cases, as described in Section 4, Table 3;C(z) is the number of road lanes;$V_{1}\left(z\right)$ and $V_{2}\left(z\right)$ are the average vehicular speeds in z-th location in the cases of uncongested and congested situations, respectively, if the specific velocity cannot be measured or reconstructed. For example, $V_{2}\left(z\right)$ is the vehicular speed in the case of a traffic congestion situation, and we assume that the value of $V_{2}\left(z\right)$ varies from 0 to 5 km/h in accordance with the road characteristics at the *z*-th location.

Therefore, according to Equation (1), for each time instant, it is possible to classify the traffic as congested or uncongested and thus to compute the contributions of CO_2_. This approach may lead to computing the whole city’s traffic flow CO_2_ production over the day, as well its distribution on the map. 

### 6.4. Assessing Seasonal Changes of Estimations: Experimental Results

Equation (1) should be satisfied for each observation of traffic flow and CO_2_ in each segment. It is possible to estimate the unknowns $K_{1}\left(z\right)$ and $K_{2}\left(z\right)$ for each location in which CO_2_ and traffic flow data are known. The estimation can be performed by means of a multilinear regression in which the dependent variable is the amount of ${\mathrm{gCO}}_{2}$ in a volumetric section of the road segment at a given time t, and the explanatory variables are the traffic count in each condition (congested and uncongested): $F_{i}\left(t,z\right)L\left(z\right)$. 

To this end, we estimated $K_{1}\left(z\right)$ and $K_{2}\left(z\right)$ for a number of sensors, and in all of them the multilinear regression turned out to be significant, producing values for the coefficients with a *p*-value in the range of *p*-value = 2 × 10^−16^ and a *t*-value > 19 in all cases for all coefficients. The R-squared of the models was also statistically significant, typically greater than 0.7 in most cases, which means that the models typically explained 70% of the variability of the response data around their mean at each hour of the day. The mean absolute percentage error (MAPE), computed as $\frac{1}{n}\;\overset{n}{\underset{i=1}{\sum }}|\frac{obs_{i}-pred_{i}}{obs_{i}}|$, is commonly used as a loss function for regression problems by means of its interpretation in terms of relative error. 

The proposed method, unlike the approaches presented in [12,13,14,15,16,17], allows the CO_2_ emission to be estimated on city roads. Following the same argument of [17], but with the aim of computing values over time and precisely in space, the results of the present work were validated in the peak traffic periods and, unlike [17], the related unknown CO_2_ emissions factors $K_{1}\left(z\right)$, $K_{2}\left(z\right)$ were computed for each timestamp, estimating the relationship between CO_2_ sensor measurements and actual traffic flow data in specific segments according to Equation (1). Figure 9 shows the model results in terms of MAPE at CO_2_ sensor locations in traffic periods when uncongested and congested traffic situations arose. The mean absolute percentage error was close to 10% for each sensor location and season. Annually, the related uncertainty was 9.1%, admitting a minimum of 5%.

The unknown emission factors $K_{1}\left(z\right)$ and $K_{2}\left(z\right)$ for each sensor location and for each period turned out to have different values in different seasons. Thus, the regressions results were computed in different seasons, taking a range of days to collect different behaviours, as described above. The summary of results in terms of emissions factor coefficients and vehicular speeds is reported in Table 4 for a number of control points. As a result, the maximum value of $K_{2}$ (amount of gCO_2_/km per car, in congested conditions) was obtained during the cold seasons, whereas the minimum value of $K_{1}$ (amount of gCO_2_/km per car in uncongested conditions) occurred during the warm seasons. This makes sense, since in winter cars tend to consume more due to the heating needs, whereas in warm conditions heating is needed less. Generally, the average use of cars increases in the cold season compared to the warm season, as in the latter citizens can choose other means of transport such as bicycles, scooters, motorbikes, etc., which reduce the polluting impact. The mean value reported in Table 4 can be considered a characterisation of traffic emissions in the network during an uncongested and congested traffic situation for the seasons. 

Moreover, the present work takes into account only traffic emissions for the purpose of the CO_2_ measured in the air. It is known that, in a real-world context, other emissions influence the total amount of CO_2_ in the air, such as, for example, natural gas consumption for domestic heating and cooking. The relationship between traffic emissions and the amount of CO_2_ is considered the net of other influences in terms of emissions. Hence, the present work should be seen as an upper boundary or the worst case of the influence of the amount of CO_2_ in the context of the transport sector in the city of Florence, where the measured CO_2_ emissions are only related to the vehicular burning of fossil fuels. From an initial comparison, it seems that the study in [14], conducted about a decade ago in the same observation area or in the immediate proximity, defined higher CO_2_ contributions in terms of emissions factors under high congested conditions with respect to the ones estimated in the present paper. This bodes well for policies to reduce the impact environmental CO_2_ combined with the evolution of vehicle engine technologies and the incremental use of electric vehicles allowing air quality to be improved gradually.

From Equation (1), the estimated (mean) values of $K_{2}$ and $K_{1}$ can be also used to compute the amount of CO_2_ at the traffic sensor locations, which are far from the air-quality sensors. This allows the amount of CO_2_ emissions to be estimated in locations where air-quality sensors are not placed. More precisely, the CO_2_ data at a given timestamp in a given period of the year can be computed by means of both the (mean) emissions factors estimated in that period (see Table 4) and the traffic data measured in that timestamp according to Equation (1). Since Florence Municipality admits many traffic sensors in scattered positions, the CO_2_ data can also be estimated in the whole area by applying interpolation methods. For example, Figure 10 shows the estimated CO_2_ emissions in the area of Florence at 8 a.m. on a spring workday by using the related traffic data at the sensors’ locations depicted in Figure 2b.

## 7. Conclusions

The emission of CO_2_ is mainly due to fossil combustion, and therefore from traffic flow (vehicle population, season, traffic behaviour, etc.), but the contribution from household heating produces a lower impact at ground level where the CO_2_ sensors are placed and where people walk. In most medium and large cities, traffic’s contribution to CO_2_ is very relevant. The total CO_2_ production of a city could be computed by taking into account the total fuel transformation in energy using several factors and efficiency aspects of the population of vehicle engines, heating, industries, etc. Moreover, the amount of CO_2_ produced by vehicles can be more specifically estimated by taking into account the weights moved, the distances, the duration of the trips and emissions factor of each specific vehicle or category, or by taking into account the distribution of vehicle types in the region or city without enabling a detailed computation of the CO_2_ in the city areas. *These approaches are unsuitable for estimating and understanding the impact of vehicles in terms of CO_2_ in the city area*, since the vehicles produce CO_2_ depending on their specific efficiency, producer, fuel, weight, driver style, road conditions, etc., which are typically different in different areas of the city, and thus also in different manners in different seasons of the year. Most cities have a large number of traffic flow sensors, whereas the number of CO_2_ sensors is limited. In the proposed model, we identified a method to determine whether vehicular traffic behaviour is congested or uncongested. This was the first step, since these two traffic situations contribute differently to the amount of CO_2_ emitted into the atmosphere. 

Therefore, we identified a method for estimating the emissions factor in different traffic conditions. This allowed us to identify a new model and method for computing CO_2_ on the basis of traffic flow data with respect to the state of the art. Therefore, the approach presented in the paper proposed, as basic elements of the solution, (i) a method for determining the emission factors, taking into account different traffic behaviours, from fluid traffic to “stop-and-go” (uncongested and congested, respectively); (ii) an analysis of changes in the emissions factors in different periods of the year; and (iii) a validation of the estimation of CO_2_ from traffic flow data. The validation was performed in the City of Florence, where a number of CO_2_ sensors and traffic flow sensors are located. In some cases, they were collocated, thus creating good conditions for validation. In order to understand the actual impact of traffic density in such locations, a traffic flow reconstruction algorithm was applied to estimate the traffic volume in certain road segments of the network, which were the closest ones to the identified air-quality sensors for validation. The proposed approach can be easily replicated in other cities to compute their emissions factors according to the characterisation of the traffic, vehicle population, and behaviour in different parts of the city and moment in time. From a long-term perspective, the assessment of emissions factors can provide useful information to verify the effective reduction of CO_2_ emissions and the impact of traffic flow due to the push towards electric vehicles. The approach should also be tested for the estimation of other pollutants produced by vehicles, first by identifying their corresponding emissions factor with the method presented in this paper. Moreover, the study could also evolve by taking into account the weather conditions, which could influence the propagation of emissions based on the wind and humidity. In the proposed model, those aspects are included in some measure in the seasonal changes of the emissions factors. 

## Figures and Tables

**Figure 1 sensors-22-03382-f001:**
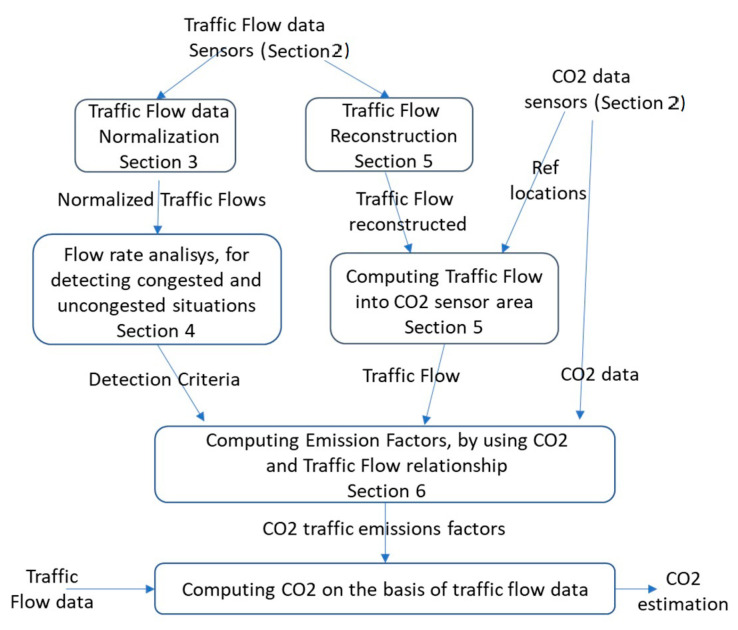
Data flows and solutions to pass from traffic flow to CO_2_ estimation, with paper section mapping.

**Figure 2 sensors-22-03382-f002:**
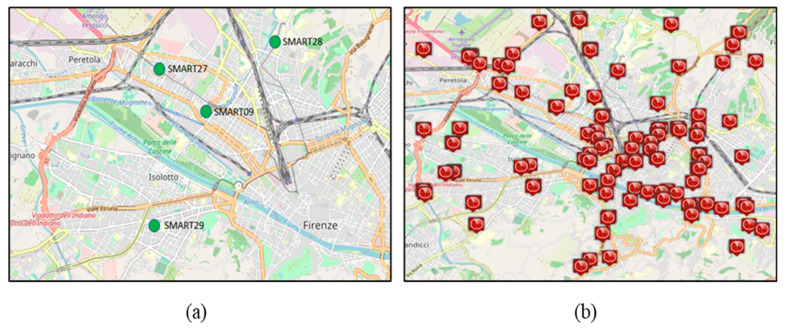
The left side depicts the locations of the air-quality sensors considered in the present work (4 of the 10 in the area of Florence). The right side depicts the map of the traffic sensors in Florence Municipality. (**a**) CO_2_ sensors; (**b**) traffic flow sensors.

**Figure 3 sensors-22-03382-f003:**
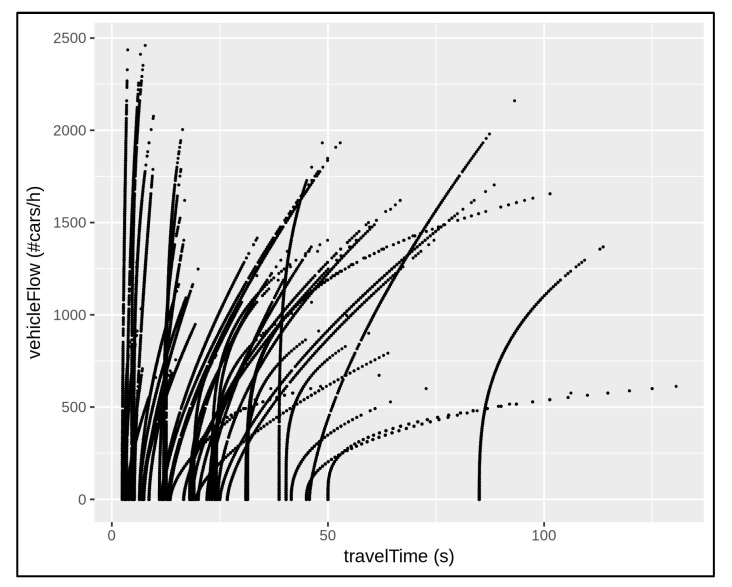
The collection of monitored traffic data coming from about 50 traffic sensors at sparse locations in the network over 7 days of observation. Each traffic sensor location presents a specific traffic behaviour depending on the travel time.

**Figure 4 sensors-22-03382-f004:**
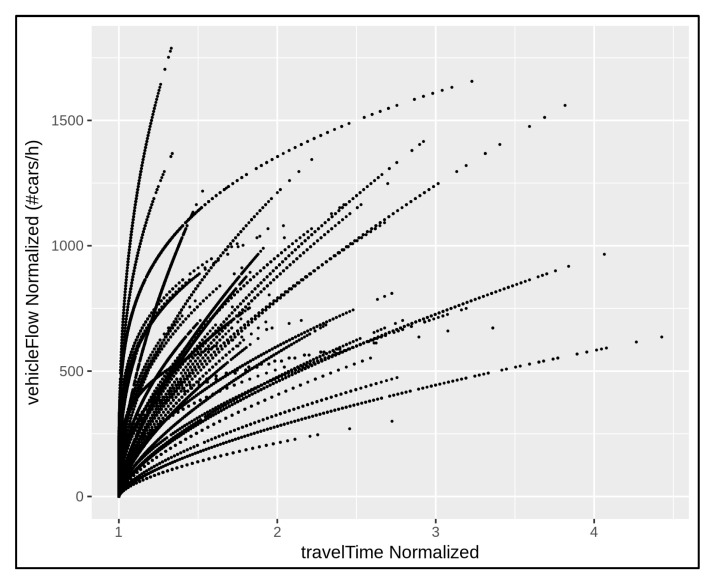
Traffic data alignment referring to the traffic trends depicted in Figure 3. It is performed by means of a normalisation approach that involves the number of road lanes, minimum travel time, and speed limit for each traffic sensor location.

**Figure 5 sensors-22-03382-f005:**
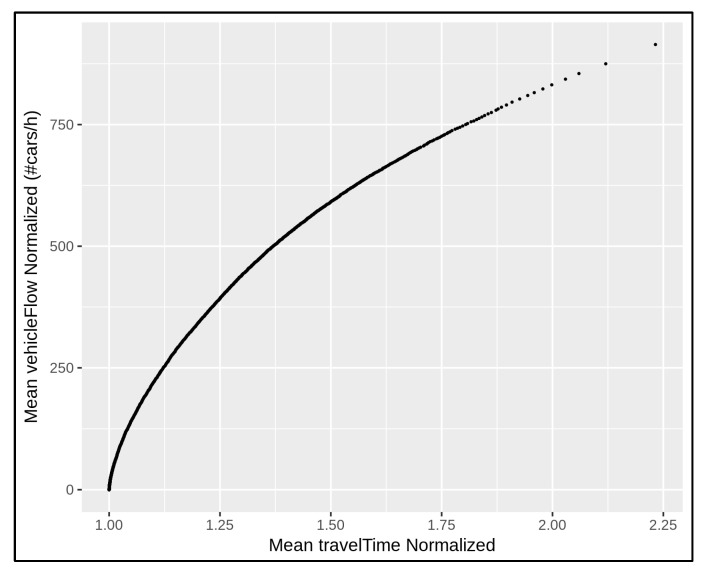
The average traffic trend that constitutes a mean behaviour by taking into account the different trends depicted in Figure 4.

**Figure 6 sensors-22-03382-f006:**
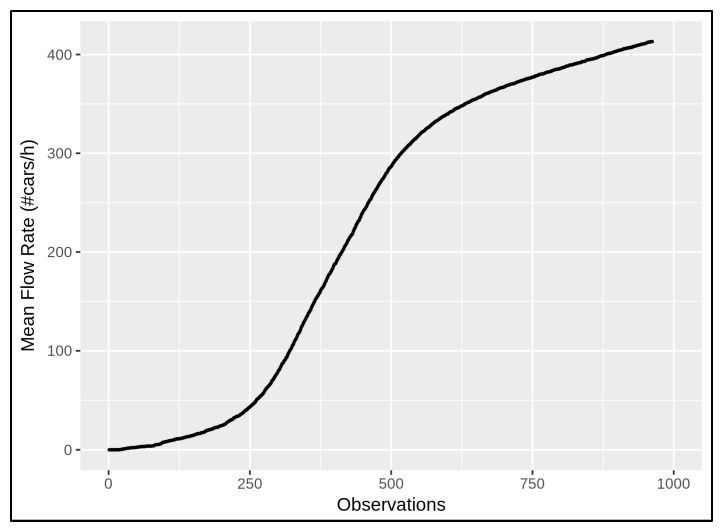
The mean flow rate (MFR) considers traffic observations that are sorted according to increasing travel time measurements in the traffic sensors depicted in Figure 5.

**Figure 7 sensors-22-03382-f007:**
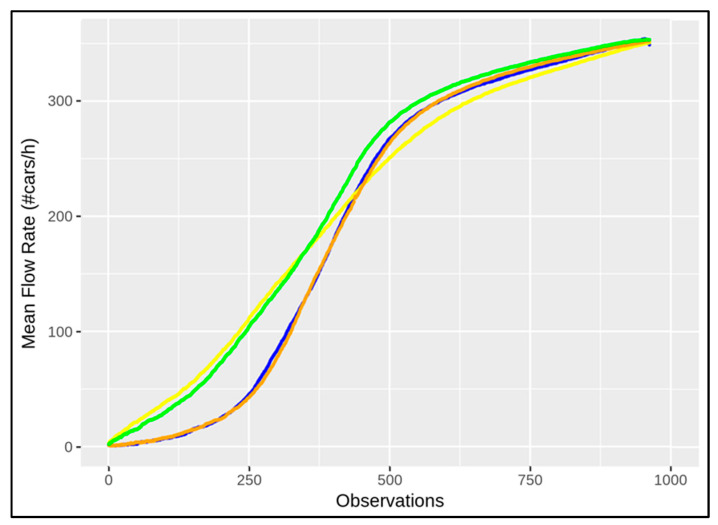
The seasonal changing curves are shown in terms of MFR, where the behaviours of the seasons are depicted in blue (March), green (May), yellow (July), and orange (October).

**Figure 8 sensors-22-03382-f008:**
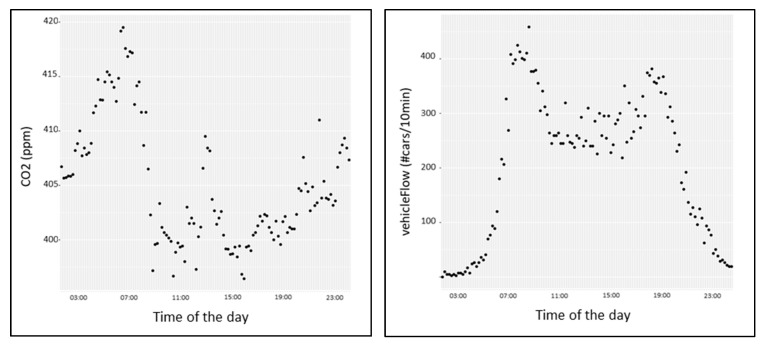
On the left side: the time alignment of CO_2_ data in a specific location, passing from ppm/2 min to ppm/10 min, at a certain time of day. On the right side: the time alignment of traffic flow data in a specific location, passing from #cars/h to #cars/10 min, at the same time of day.

**Figure 9 sensors-22-03382-f009:**
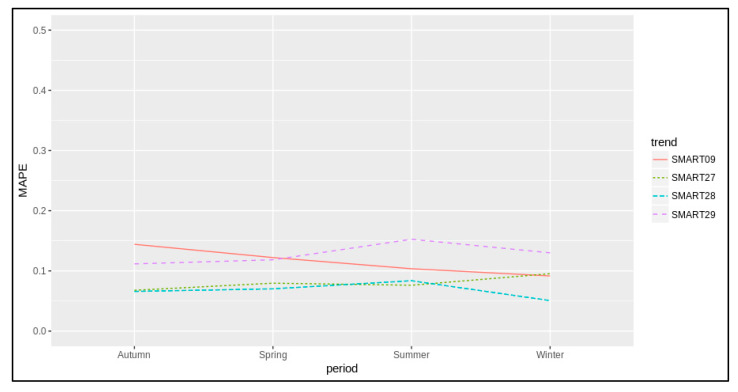
Multilinear regression model results in terms of MAPE according to each sensor location and each period of the year. The results were obtained during the peak traffic periods when uncongested and congested traffic situations arise and significant CO_2_ emissions occur.

**Figure 10 sensors-22-03382-f010:**
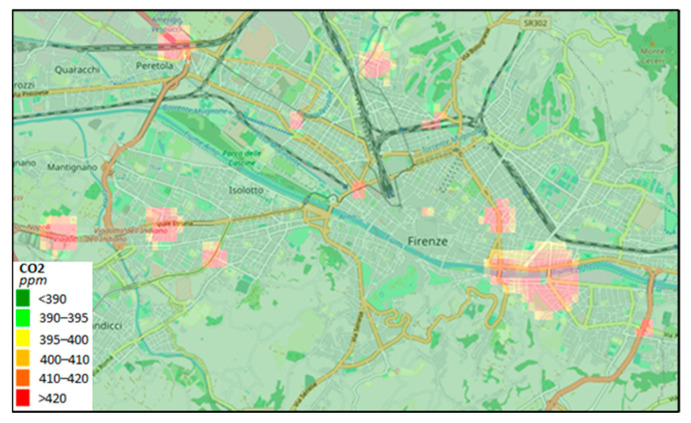
An example of a CO_2_ heatmap in Florence Municipality. An interpolation method is applied to the CO_2_ data, which are estimated in accordance with Equation (1) at the traffic sensor locations depicted in Figure 2b. The traffic data are simultaneously measured in sparse locations at 8 a.m. on a spring workday. Accessible on Snap4city.org https://www.snap4city.org/dashboardSmartCity/view/index.php?iddasboard=MTUzMg== (accessed on 27 April 2022).

**Table 1 sensors-22-03382-t001:** Numerical values coming from the seasonal changing curves in terms of mean flow rate depicted in Figure 7.

Mean Flow Rate	100	250	400	550	700	850	950
*October*	7.3	45.1	178.3	289.0	320.0	340.4	353.7
*July*	38.0	110.9	197.3	271.4	312.2	335.6	349.4
abs. deviation	30.7	65.8	19.0	17.5	7.7	4.8	4.2

**Table 2 sensors-22-03382-t002:** An example of the coefficients related to the polynomial approximations of the MFR(Spring) and FR(Spring,i) for 3 traffic flow sensors during the (same) selected period of time in spring.

Polynomial Approx.	a	b	c	d
*MFR(Spring)*	−0.0000011	0.0014465	0.0686048	−17.38807
*FR(Spring,1)*	−0.0000019	0.0027314	−0.140624	−9.454277
*FR(Spring,2)*	−0.000001	0.0016093	−0.247455	2.96993
*FR(Spring,3)*	−0.0000019	0.002776	−0.471167	11.87328

**Table 3 sensors-22-03382-t003:** An example of the coefficients related to the polynomial approximations of the MFR(Spring) and FR(Spring,i) for 3 traffic flow sensors during the (same) selected period of time in spring.

Traffic Flow Sensors	Inflection Point (T-TH OBS) on P	Flow Rate (#Cars/h)	Traffic Flow (#Cars/h)	Traffic Density (#Cars/Km)
*All city sensors*	438	215.8	255	7.24
*S1 (near SMART27)*	479	366	366.88	7.12
*S2 (near SMART28)*	536	186.14	216	6.35
*S3 (near SMART29)*	487	234.24	240	4.21

**Table 4 sensors-22-03382-t004:** Summarising numerical results by means of the applied model to identify the desired emissions factors in case of uncongested and congested traffic situations in different periods of the year for different sensor locations.

	**Winter**				**Autumn**			
**Air Sensor**	$K_{1}$	$K_{2}$	$V_{1}$	$V_{2}$	$K_{1}$	$K_{2}$	$V_{1}$	$V_{2}$
** *SMART09* **	230.0	681.3	37.4	3.9	317.7	791.5	36.5	3.9
** *SMART27* **	160.8	349.7	46.0	1.0	161.7	321.7	44.0	1.0
** *SMART28* **	219.6	386.7	36.0	1.0	253.1	352.3	35.0	1.0
** *SMART29* **	296.0	732.0	35.1	1.5	355.5	520.0	53.9	1.5
** *MEAN* **	226.6	537.4	38.6	1.8	272.0	496.3	42.3	1.8
	**Summer**				**Spring**			
**Air Sensor**	$K_{1}$	$K_{2}$	$V_{1}$	$V_{2}$	$K_{1}$	$K_{2}$	$V_{1}$	$V_{2}$
** *SMART09* **	184.0	709.4	39.8	3.9	217.8	619.2	35.2	3.9
** *SMART27* **	133.6	323.3	53.2	1.0	150.4	317.5	51.5	1.0
** *SMART28* **	290.3	383.2	36	1.0	274.0	381.5	34.0	1.0
** *SMART29* **	264.5	643.7	46.9	1.5	315.3	589.6	57.0	1.5
** *MEAN* **	218.1	514.9	43.9	1.8	239.3	476.9	44.4	1.8

## Data Availability

Data accessible on the Snap4City platform. Available online: https://www.snap4city.org (accessed on 27 April 2022).

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
