# Peer review of "Estimating CO2 Emissions from IoT Traffic Flow Sensors and Reconstruction"

_sensors, 2022, doi:10.3390/s22093382_

Round 1

Reviewer 1 Report

The authors propose a model to estimate the CO2 emissions from traffic flow data and focusing on uncongested and congested conditions. The research is based on data extracted from sensors (in terms of traffic information and CO2 concentration) and makes use of techniques for traffic flow reconstruction.

The study is very relevant and gives promising insights regarding how technology can help policy makers and traffic planners in designing strategies for minimising GHG-type effects.

The manuscript is very well organised and the sections provide, in general, enough information for the readers.

I would just recommend to split the first block of text into a brief context introduction (Introduction) and a critical description of state-of-the-art niche studies (Literature Review).

The only drawback seems to be related to the possibility of scalability of such framework in case no pollutant emission sensors would be available. I mean, if we would like to estimate CO2 based on some especial models. The thing will be more complicated in case of NOx assessment - the sensors on traffic information appear to provide information on traffic flow like average speed in the covered sensor spot, and in this case, microscopic information would be crucial to account for harsh accelerations, for instance. Maybe the authors can include some limitation regarding this type of evaluations. This is particularly relevant, since most of pollutant emission sensors require much more maintenance efforts (also in economic terms) than traffic sensors.

Maybe I missed, but the multilinear regression model equation should be clearly presented. What are in fact the explanatory variables?

Please, simplify the caption of TABLE IV.

L685: Avoid sentences like "The proposed approach proposed".

Author Response

The authors propose a model to estimate the CO2 emissions from traffic flow data and focusing on uncongested and congested conditions. The research is based on data extracted from sensors (in terms of traffic information and CO2 concentration) and makes use of techniques for traffic flow reconstruction.

The study is very relevant and gives promising insights regarding how technology can help policy makers and traffic planners in designing strategies for minimising GHG-type effects.

ANSWER: thanks.

The manuscript is very well organised and the sections provide, in general, enough information for the readers.

ANSWER: thanks.

I would just recommend to split the first block of text into a brief context introduction (Introduction) and a critical description of state-of-the-art niche studies (Literature Review).

ANSWER: The introduction has been split into “Introduction” and “Literature review”.

The only drawback seems to be related to the possibility of scalability of such framework in case no pollutant emission sensors would be available. I mean, if we would like to estimate CO2 based on some especial models.

ANSWER: As we added I the new version of the paper, the presence of almost collocated CO2 and of traffic flow sensors allowed us to compute the specific emission factors. On the other hand, their absence, or their strong reduction could be overcome with the usage of rented CO2 sensors or to use the typical values of the emission factors in the area, vanishing the precision of the measure.

The thing will be more complicated in case of NOx assessment - the sensors on traffic information appear to provide information on traffic flow like average speed in the covered sensor spot, and in this case, microscopic information would be crucial to account for harsh accelerations, for instance. Maybe the authors can include some limitation regarding this type of evaluations. This is particularly relevant, since most of pollutant emission sensors require much more maintenance efforts (also in economic terms) than traffic sensors.

ANSWER: To address the NO2 estimation from traffic was not in our goals. We added in the conclusion a statement. The approach should be also tested for the estimation of other pollutants also produced by the vehicles.

Maybe I missed, but the multilinear regression model equation should be clearly presented. What are in fact the explanatory variables?

ANSWER: It has been clarified in Section 6.4

Please, simplify the caption of TABLE IV.

ANSWER: It has been implemented I the new version of the paper.

L685: Avoid sentences like "The proposed approach proposed".

ANSWER: thank, solved!

Reviewer 2 Report

The research is considered relevant in order to identify the impact of CO2 vehicle emissions. There are some aspects to improve, which are detailed below:
1.
The title seems fine to me, as well as the abstract. However, at the end of
the abstract it is important to present what the relevant results of the
research were, rather than to which project it belongs.
2. Throughout the document, in all the proposed sections there are paragraphs
that are too long, which make it difficult to adequately reading. Improvement
is required on this matter, as the current paragraphs mention different topics.

3. There are some words in bold that I think it is not appropriate to present
them in this way, for example: "In this paper", "The paper is structured as
follows."

4. Although in the penultimate paragraph of the introduction an attempt is made
to present the current situation of the literature (regarding to the estimate of
CO2 emissions from vehicles, based on traffic data) and what is the difference
with respect to proposed research. It is not possible to adequately summarize
this aspect, several things are mentioned, but it is difficult identify what is
most relevant. It is recommended to improve this aspect.

5. I think that sections 2 to 6 should be much better structured.
I think it is very important to differentiate the methods used and
the results obtained in all work. As the information is divided into
sections 2 to 6 it is not well understood that corresponds to methods,
tests and experiments carried out by the authors and corresponds to results.
It is recommended greatly improve the organization.

6. There is a section 5.1., but there is no other section at this level.
There is no section 5.2, for example. It is recommended avoid this type
of numbering, in the final organization of the document they make,
as required in the previous point.

7. Regarding section 7 of conclusions, most (from the beginning to the middle
approximately) of said section repeats what mentioned in the abstract, it is
recommended to modify this section and improve it considerably.

Author Response

The research is considered relevant in order to identify the impact of CO2 vehicle emissions. There are some aspects to improve, which are detailed below:

  1. The title seems fine to me, as well as the abstract. However, at the end of the abstract it is important to present what the relevant results of the research were, rather than to which project it belongs.

ANSWER: thanks, it has bee corrected.

  1. Throughout the document, in all the proposed sections there are paragraphs that are too long, which make it difficult to adequately reading. Improvement is required on this matter, as the current paragraphs mention different topics.

    ANSWER: The paper has been revised and improved in the direction of solving the problem as much as possible.
  2. There are some words in bold that I think it is not appropriate to present them in this way, for example: "In this paper", "The paper is structured as follows."

    ANSWER: thanks, it has bee corrected.
  3. Although in the penultimate paragraph of the introduction an attempt is made to present the current situation of the literature (regarding to the estimate of CO2 emissions from vehicles, based on traffic data) and what is the difference with respect to proposed research. It is not possible to adequately summarize
    this aspect, several things are mentioned, but it is difficult identify what is most relevant. It is recommended to improve this aspect.

    ANSWER: the introduction has been structured according to your comment and to the comment of reviewer 1.
  4. I think that sections 2 to 6 should be much better structured.
    I think it is very important to differentiate the methods used and the results obtained in all work. As the information is divided into sections 2 to 6 it is not well understood that corresponds to methods, tests and experiments carried out by the authors and corresponds to results. It is recommended greatly improve the organization.

ANSWER: The paper organization has been better explained in Section 1, in which Figure 1 has been included to describe the paper structure with respect to the data flows and algorithms put in place and described in Sections 3, 4, 5 and 6. So that, with this figure the reader can identify at a glance the motivations for paper structure and general view. In particular the method is described in Sections 3,4,5 and 6, while the experimental results are closed in Section 6.4.

  1. There is a section 5.1., but there is no other section at this level.
    There is no section 5.2, for example. It is recommended avoid this type of numbering, in the final organization of the document they make, as required in the previous point.

    ANSWER: done
  2. Regarding section 7 of conclusions, most (from the beginning to the middle approximately) of said section repeats what mentioned in the abstract, it is recommended to modify this section and improve it considerably.

ANSWER: Thanks a lot, the Section on conclusions has be strongly revised.

Reviewer 3 Report

[Comment 1] Novelty

[Subcomment 1a] (lines 112-116) I believe these two statements are related to the novelty of this study. For clarity, the authors must mention specifically, what is newly considered in their study and how each new aspect differs from which specific previous study.

[Subcomment 1b] Please clearly state the contributions of this study for clarity.

[Subcomment 1c] When comparing the novelty aspects with previous studies, I suggest the authors present the comparison in a table for clarity.

[Subcomment 1d] (Section 6.3) Did the authors develop all equations in the manuscript or did the authors refer to any previous study? Please state it clearly in the manuscript.

[Subcomment 1e] (page 17) How does this study differs from References [32] and [33]?

[Comment 2] Methodology

[Subcomment 2a] (lines 245-255) For clarity, please provide the detailed formulas to conduct the normalization.

[Subcomment 2b] Please provide some calculation examples (before and after the normalization) to show that the normalization is necessary.

[Subcomment 2c] (lines 415-416) The explanation is unclear. The readers might misunderstand that the authors calculate the traffic volume and use it to estimate the amount of CO2, instead of just calculating the CO2 amount using sensors directly.

[Subcomment 2d] I understand that the authors separately calculate the traffic data and the CO2 data, then use the traffic data to modify the CO2 data measurement. Is it true? I could not find any specific explanations anywhere in the manuscript, that makes it difficult for me to evaluate the study. My point is, the authors need to explain every research step in detail and provide justifications why they did it (e.g., by presenting supporting references).

[Subcomment 2e] Please explain the data collection and processing framework in a clear figure in any earlier section, for clarity.

[Comment 3] Writing quality and clarity

[Subcomment 3a] In general, the background for the research methodology is unclear. The authors need to appropriately address their proposed methodology and support any conducted calculations with appropriate reasonings and references. I believe there might be some good explanations, but I could not see them presented well, so I conclude that the presentation is very poor and difficult to be understood.

[Subcomment 3b] (Abstract) Did the authors mean 100 number of sensors or 100 types of sensors?

The authors need to explain what type of sensors used and types of data measured in their study (by each sensor).

[Subcomment 3c] Please revise Figures 1ab into Figures 1(a) and 1(b).

[Subcomment 3d] (Figure 6) Please state which month represent what season in detail.

[Subcomment 3e] Please use a consistent line spacing throughout the manuscript.

Author Response

[Comment 1] Novelty

[Subcomment 1a] (lines 112-116) I believe these two statements are related to the novelty of this study. For clarity, the authors must mention specifically, what is newly considered in their study and how each new aspect differs from which specific previous study.

ANSWER: In Section 1, we clarified that the proposed approach

  • is new with respect to the state of the art since it provides a method for computing CO2 emissions locally in the city, and not only for specific vehicle or globally at city level.
  • allows to estimate the CO2 produced by vehicles in the city, which is a relevant contribution of CO2 produced in the city, directly measuring the impact of vehicle population on the production of CO2, thus increasing the precision in measuring CO2. The advantage to identify the function from traffic flow to CO2 may be used to estimate the CO2 in city areas by knowing the traffic flow with a certain precision, and thus the total CO2 production of the city, which is presently only coarsely guessed by using a very limited number of CO2 sensors, while most of the cities have hundreds of traffic flow sensors.
  • is based on: (i) the identification of the relationships from the measured traffic flow and determining the emission factors taking into account different traffic behaviours from fluid traffic to “stop-and-go” conditions (congested and uncongested traffic situations), (ii) the changes of the emission factor in different periods of the year, (iii) an approach for statistical validation by means of the CO2 measures taken from specific sensors and traffic flow data, allowed the assessment of precision of the indirect estimation of CO2 on the basis of traffic flow.

[Subcomment 1b] Please clearly state the contributions of this study for clarity.

ANSWER: In Section 1, we clarified that the proposed approach.

[Subcomment 1c] When comparing the novelty aspects with previous studies, I suggest the authors present the comparison in a table for clarity.

ANSWER: Since the other approaches are very different the comparison in terms of table is not actually viable.

[Subcomment 1d] (Section 6.3) Did the authors develop all equations in the manuscript or did the authors refer to any previous study? Please state it clearly in the manuscript.

ANSWER: Equation (1) has been derived from the physical analysis of the phenomena, while the decomposition of terms of congested and uncongested contributions from traffic as described and identified in the paper according to section 4 is totally new. It has been clarified in Section 6.3 of the new version of the paper.

[Subcomment 1e] (page 17) How does this study differs from References [32] and [33]?

ANSWER: As explained in the paper, section 6.4, in [32] the estimation of CO2 is performed street by street as a Total amount produced annually, making an hypothesis amount the composition of the fleet of vehicles by [33] in the area and without assessing the actual presence of congested and uncongested situations, over time and by road segment. In our case, the estimation if punctual over time and space, and this approach has been possible on the basis of the classification approach provided in Section 4, and thus it enabled us to define the solution in Section 6.

[Comment 2] Methodology

[Subcomment 2a] (lines 245-255) For clarity, please provide the detailed formulas to conduct the normalization.

ANSWER: In Section 3 of the new version of the paper, the formulas adopted for the normalization have been provided.

[Subcomment 2b] Please provide some calculation examples (before and after the normalization) to show that the normalization is necessary.

ANSWER: In Section 3 of the new version of the paper, some examples regarding the normalization have been provided.

[Subcomment 2c] (lines 415-416) The explanation is unclear. The readers might misunderstand that the authors calculate the traffic volume and use it to estimate the amount of CO2, instead of just calculating the CO2 amount using sensors directly.

ANSWER: thanks a lot it has been clarified in the paper.

[Subcomment 2d] I understand that the authors separately calculate the traffic data and the CO2 data, then use the traffic data to modify the CO2 data measurement. Is it true? I could not find any specific explanations anywhere in the manuscript, that makes it difficult for me to evaluate the study. My point is, the authors need to explain every research step in detail and provide justifications why they did it (e.g., by presenting supporting references).

ANSWER: In the paper, section 5, it has been clarified that the aim of the paper is to compute CO2 values on the basis of Traffic Flow data. In the new Figure 1, the flow is presented to clarify the approach. In substance, we have identified a model (see Section 6, equation (1)) by which the computation of CO2 from traffic is possible. The computation of CO2 depends on the traffic flow and of the emission factors which are different for congested and uncongested traffic flow cases. Therefore, we have found a way to detect the conditions to compute the number of vehicles passing close to the CO2 control point in the different conditions, Section 4. This approach allow us to demonstrate (validate) the model by computing the emission factors.

[Subcomment 2e] Please explain the data collection and processing framework in a clear figure in any earlier section, for clarity.

ANSWER: In the new Figure 1, the flow is presented to clarify the approach.

[Comment 3] Writing quality and clarity

[Subcomment 3a] In general, the background for the research methodology is unclear. The authors need to appropriately address their proposed methodology and support any conducted calculations with appropriate reasonings and references. I believe there might be some good explanations, but I could not see them presented well, so I conclude that the presentation is very poor and difficult to be understood.

ANSWER: The state of the art has been clarified into Section 1.1, and recalled in the validation. The approach proposed is actually new, so t hat the amount of work specially oriented on computing CO2 from traffic has been reported and commented.

[Subcomment 3b] (Abstract) Did the authors mean 100 number of sensors or 100 types of sensors?
The authors need to explain what type of sensors used and types of data measured in their study (by each sensor).

ANSWER: As described in the paper, in Florence there are more than 100 traffic flow sensors. They are spire and virtual spire sensors producing values of counting, traffic density and thus local traffic flows. Virtual  spire at practically TVcameras implementing the classical spire in ground measuring the flow. All of them are well calibrated and produce coherent results.

[Subcomment 3c] Please revise Figures 1ab into Figures 1(a) and 1(b).

ANSWER: done

[Subcomment 3d] (Figure 6) Please state which month represent what season in detail.

ANSWER: In the  new version of the paper, the figure 6 become Figure 7. The months have been specified.

[Subcomment 3e] Please use a consistent line spacing throughout the manuscript.

ANSWER: thanks, revised.

Reviewer 4 Report

The paper addresses a current problem of air pollution, pollution due to road traffic in large urban areas, respectively Florence.

What is represents SMART09, SMART27, SMART28 and SMART29 in this article?

The amount of pollutant emissions, especially CO2 is estimated based on traffic flow, traffic speed, road infrastructure, etc., but does not take into account the meteorological data in the area of ​​measurements that could influence these estimates.

The work is of interest for the study of air quality in urban congested areas

Author Response

The paper addresses a current problem of air pollution, pollution due to road traffic in large urban areas, respectively Florence.

ANSWER: thanks.

What is represents SMART09, SMART27, SMART28 and SMART29 in this article?

ANSWER: As described in Section 2, they are the IDs of the sensors. Those sensors are not all in critical locations for the traffic neither for pollutant SMART28 and SMART29 are in dense traffic roads, while SMAR27 and SMART09 are in mid range traffic areas.

The amount of pollutant emissions, especially CO2 is estimated based on traffic flow, traffic speed, road infrastructure, etc., but does not take into account the meteorological data in the area of ​​measurements that could influence these estimates.

ANSWER: thanks a lot for the comment. In effect taking into account the weather conditions which could influence the propagation of emission according to the wind and humidity.  We added this comment into the conclusions as a possible future work.

The work is of interest for the study of air quality in urban congested areas

ANSWER: thanks.

Round 2

Reviewer 2 Report

Some aspects of the document have been improved, according to the recommendations of the reviewers. However, I consider that there are still some details to improve, which are detailed below.

1. I consider that the differentiation of what has been done in this degree project,
with respect to the reviewed literature, is not yet sufficiently clear and summarized.

2.
A section 1.1 was created, but there is no section 1.2. 3. I think the conclusions section should be improved, since it continues to be
presented as a summary of the article, rather than what was concluded after carrying
out what was proposed.

4. Personally, with the adjustments made to the sections, I think that the article
still does not have a good organization of the information. And it is not yet clear
what corresponds to the methods used in the development of the article.

Author Response

Some aspects of the document have been improved, according to the recommendations of the reviewers. However, I consider that there are still some details to improve, which are detailed below.

1. I consider that the differentiation of what has been done in this degree project,
with respect to the reviewed literature, is not yet sufficiently clear and summarized.

ANSWER: Section 1.2 has been provided in which a clear description of the differences with respect to the state of the art solutions reported in Section 1.1 are reported with specific citations to paper [12], [13], [14], [15]. [16] and [17]. They substantially.  As reported in the analysis, the literature at the state of the art estimates the CO2 in (i) controlled experiments for each single vehicle kind as in [12], [13], [14], [15], (ii) general approach on the basis of the energy consumed (fuel) by taking into account of vehicle distribution as in [16], [17].

  1. A section 1.1 was created, but there is no section 1.2.

ANSWER: Section 1.2 has been provided in which a clear description of the differences with respect to the state of the art solutions reported in Section 1.1 are reported with specific citations to paper [12], [13], [14], [15]. [16] and [17].

  1. I think the conclusions section should be improved, since it continues to be
    presented as a summary of the article, rather than what was concluded after carrying
    out what was proposed.

ANSWER: conclusions have been improved to provide more comments on obtained results and stressing the differences.

4. Personally, with the adjustments made to the sections, I think that the article
still does not have a good organization of the information. And it is not yet clear
what corresponds to the methods used in the development of the article.

ANSWER: the articles structure has been better described into Section 1.2. To this end there is a figure that linearly describes the paper structure as a sort of data flow, putting in evidence the results produced by each section and the top addressed in the general flow and aim. I hope that this would be enough to guide the readers.  

Reviewer 3 Report

For clarity, about the novelty statement, I still suggest the authors to mention specifically, what is newly considered in their study and how each new aspect differs from which specific previous study, e.g., our study considers ... while Reference [] only considered ....

Author Response

ANSWER: Section 1.2 has been provided in which a clear description of the differences with respect to the state of the art solutions reported in Section 1.1 are reported with specific citations to paper [12], [13], [14], [15]. [16] and [17]. They substantially.  As reported in the analysis, the literature at the state of the art estimates the CO2 in (i) controlled experiments for each single vehicle kind as in [12], [13], [14], [15], (ii) general approach on the basis of the energy consumed (fuel) by taking into account of vehicle distribution as in [16], [17].